# Trivalent and Pentavalent Antimonials Impair Cardiac Mitochondrial Function in Mice

**DOI:** 10.3390/ijms26189073

**Published:** 2025-09-18

**Authors:** Itanna Isis Araujo de Souza, Maria Eduarda Maciel Fernandes Pavarino, César Francisco Maricato da Rosa, Laís Eduardo Marinho, Caroline da Silva Moraes, José Hamilton Matheus Nascimento, Antonio Carlos Campos de Carvalho, Leonardo Maciel

**Affiliations:** 1Instituto de Biofísica Carlos Chagas Filho, Universidade Federal do Rio de Janeiro, Rio de Janeiro 21941-902, RJ, Braziljhmn@biof.ufrj.br (J.H.M.N.); acarlos@biof.ufrj.br (A.C.C.d.C.); 2Programa de Pós-Graduação em Cardiologia, Universidade Federal do Rio de Janeiro, Rio de Janeiro 21941-617, RJ, Brazil; 3Campus Professor Geraldo Cidade, Universidade Federal do Rio de Janeiro, Duque de Caxias 25240-005, RJ, Brazil

**Keywords:** mitochondria toxicity, sodium antimoniate, leishmaniasis

## Abstract

Pentavalent sodium antimoniate (Sb(V)) has been used for over 50 years in leishmaniasis treatment. Sb(V) is converted into trivalent antimony (Sb(III)) within macrophages acting as a prodrug by disrupting fatty acid beta-oxidation and glycolysis, impairing the energy metabolism of the parasite. Despite extensive use, the effects of antimonials on host mitochondria are not well understood. This study investigated the impact of Sb(V) and Sb(III) on mitochondria isolated from mouse hearts via differential centrifugation and lastly incubated with Sb(V) or Sb(III). Mitochondrial function was evaluated by oxygen consumption, ATP production, reactive oxygen species (ROS) generation, and transmembrane potential. Both Sb(V) and Sb(III) reduced oxygen consumption in complex I respiratory states 1, 2, and 3 at 1 µg/mL and 1 ng/mL. ROS production increased in Sb(V)-treated mitochondria. ATP production was impaired by both drugs starting at 1 ng/mL. Proton leak also increased, and significant changes in transmembrane potential were observed at both concentrations. These findings indicate that Sb(V) and Sb(III) directly compromise mitochondrial function from isolated mouse heart mitochondria by reduced ATP production and increased ROS.

## 1. Introduction

Leishmaniasis is a group of diseases caused by various species of unicellular protozoa from the genus *Leishmania* sp., characterized by a digenetic life cycle that alternates between vertebrate hosts and insect vectors [1]. It manifests through several morphologically similar species, affecting humans. The disease is endemic in approximately 98 countries, placing around 1 billion people at risk of contracting one of its various forms [2]. In developing countries, patients with visceral or cutaneous leishmaniasis encounter significant logistical barriers to accessing treatment. Factors such as the considerable distance to healthcare facilities, lack of adequate transportation, challenges in adhering to treatment protocols, and high treatment costs contribute to poor treatment adherence [3].

Leishmaniasis can be classified into two main forms: cutaneous leishmaniasis, the most common, and visceral leishmaniasis, the most aggressive and life-threatening when left untreated [4]. Leishmaniasis has a complex life cycle, with one of its developmental stages, the amastigote, residing within the host’s immune cells. This intracellular localization complicates the ability of drugs to effectively target the parasites [5]. The primary goal of treatment is to eliminate the intracellular parasites, making chemotherapy the most effective therapeutic approach. Chemotherapy remains the cornerstone of disease management [6].

For over a century, pentavalent antimonials have been the first-line treatment for leishmaniasis [7]. However, their exact mechanism of action remains poorly understood. It is still unclear whether the active antimonial form is pentavalent (Sb(V)) or trivalent (Sb(III)) [8,9,10]. Sb(V) is generally considered a prodrug that requires intracellular reduction to Sb(III) for activation, the latter being responsible for its antiparasitic effects. For antimony to exert a leishmanicidal action, Sb(V) must penetrate the host macrophage, cross the phagolysosomal membrane, and reach the parasite intracellularly [11]. The pro-drug model proposes that Sb(V) is reduced to Sb(III) by thiol-containing molecules such as cysteine (Cys) and cysteinyl-glycine (Cys-Gly) in the host’s lysosomes and by trypanothione (T(SH)_2_) in the parasite [11]. In addition, parasite-specific enzymes, such as thiol-dependent reductase (TDR1) and antimonial reductase (ACR2), are implicated in the bioactivation of Sb(V) [12,13,14].

Once reduced, Sb(III) forms stable 1:1 and 1:3 complexes with intracellular thiols such as glutathione (GSH) and trypanothione (T(SH)_2_), inhibiting trypanothione reductase and leading to redox imbalance and oxidative stress [15,16]. This oxidative stress is manifested by lipid peroxidation, a decrease in glutathione peroxidase activity, and the release of lactate dehydrogenase [17]. Sb(III) also inhibits essential metabolic enzymes such as succinyl dehydrogenase and phosphofructokinase, disrupting energy production in the parasite [18]. These Sb(III)-thiol complexes can be sequestered in the vacuoles or exported via ABC transporters [19,20]. Increased intracellular levels of thiols (Cys, GSH, T(SH)_2_), overexpression of genes involved in GSH and polyamine biosynthesis, and up-regulation of ABC transporters have been associated with antimonial resistance [20]. In particular, inhibition of GSH biosynthesis was able to reverse resistance to Sb(V) in isolates of *Leishmania donovani* in vivo, suggesting a potential therapeutic strategy through combined treatments [21]. This conversion occurs intracellularly in macrophages [11]. Once activated, Sb(III) interrupts the β-oxidation of fatty acids and the glycolysis of the parasite, further contributing to the elimination of the parasite [22].

Despite their efficacy, the drugs used to treat leishmaniasis present significant drawbacks, including high toxicity and numerous adverse effects, often resulting in treatment abandonment and the emergence of drug-resistant strains [22,23]. Although treatment with Sb(V) has resulted in fewer patient refusals, similar cardiovascular effects to those seen with trivalent antimonials, such as hypotension, prolonged QT intervals, ST-segment changes, and severe arrhythmias (e.g., Torsades de Pointes), have been observed, which can lead to sudden death [24]. These cardiotoxic effects could be thought to result from the conversion of Sb(V) to Sb(III), which preserves certain toxic properties [16]. It is believed that these effects are dose- and duration-dependent [25], associating the cardiac damage to oxidative stress, lipid peroxidation, reduced glutathione (GSH) levels, inhibition of glutathione peroxidase, alterations in cellular thiols, and the release of lactate dehydrogenase (LDH) [26].

Oxidative stress caused by redox cycling or depletion of antioxidants such as glutathione can also impair mitochondrial function [27]. Given the structural and functional complexity of mitochondria, it is not surprising that numerous mechanisms could lead to drug-induced mitochondrial toxicity [28]. The exact incidence of mitochondrial respiratory chain dysfunction caused by drug-induced toxicity has yet to be fully determined [28].

In addition, previous studies have shown that compounds containing antimony can induce mitochondrial dysfunction in various cell types, reinforcing the relevance of investigating these effects in the cardiac context. Losler [29] observed that antimony trioxide (Sb_2_O_3_) promotes apoptosis and mitochondrial membrane potential (Δψm) depolarization in hematopoietic cell lines, effects that are potentiated by depletion of intracellular glutathione with buthionine sulfoximine (BSO). Verdugo [30], in turn, demonstrated that Sb(III) compromised mitochondrial respiratory activity in HEK-293 cells and induced protein aggregation, as evidenced by reduced protein mobility in FRAP experiments. More recently, Lou et al. [31] reported that exposure to Sb promoted the growth of bladder tumor cells by reducing Δψm, inhibiting the activity of complexes I-IV of the respiratory chain, and blocking the PINK1/Parking-mediated mitophagy pathway. These findings together show that the deleterious effects of antimony on mitochondria are not restricted to a single cell type and contribute to understanding its toxic potential in tissues with a high bioenergetic demand, such as the myocardium.

This study is justified by the pressing need to deepen our understanding of how antimonial drugs impact mitochondrial bioenergetics, as well as their structure and function. Therefore, here, we investigate the effects of Sb(V) and Sb(III) on the function of isolated mitochondria from hearts.

## 2. Results

### 2.1. Mitochondrial Oxygen Consumption

The oxygen consumption of isolated mitochondria was analyzed following exposure to Sb(V) at doses of 1 ng/mL and 1 µg/mL in mouse hearts (Figure 1). In state 1 of complex I, mitochondrial oxygen consumption was reduced in both the 1 ng/mL and 1 µg/mL Sb(V) groups compared to the control group (Figure 1A). Similarly, in state 2 of complex I, where oxygen consumption was stimulated by glutamate and malate, a reduction was observed at the 1 µg/mL dose compared to the control group (Figure 1B). In state 3 of complex I, oxygen consumption was lower in both the 1 ng/mL and 1 µg/mL doses of the Sb(V) group relative to the control (Figure 1C). No statistically significant differences were observed in mitochondrial respiration through complex IV (Figure 1D) or in uncoupled maximal oxygen consumption (Figure 1E), suggesting similar mitochondrial loading controls and viability, respectively, across groups.

Exposure of mitochondria to Sb(III) at 1 ng/mL and 1 µg/mL also affected oxygen consumption (Figure 2). In state 1 of complex I, respiration was lower in both treatment groups compared to the control (Figure 2A). In state 2 of complex I, oxygen consumption stimulated by glutamate and malate was similarly reduced at both dosages (Figure 2B). In state 3 of complex I, oxygen consumption was significantly decreased at both the 1 ng/mL and 1 µg/mL Sb(III) doses compared to the control group (Figure 2C). Similar to Sb(V), no statistically significant differences were observed in mitochondrial respiration through complex IV (Figure 2D) or in uncoupled maximal oxygen consumption (Figure 2E), suggesting similar mitochondrial loading controls and viability, respectively, across groups.

### 2.2. Mitochondrial ROS Production and ATP Production

Mitochondrial reactive oxygen species (ROS) production was significantly elevated in the presence of Sb(V) at both 1 ng/mL and 1 µg/mL doses compared to the control group (Figure 3A). However, no differences in ROS production were observed following exposure to Sb(III) at either dose (Figure 3B). Mitochondrial ATP production decreased in mitochondria treated with Sb(V) at both dosages (Figure 3C). Similarly, a reduction in ATP production was noted following exposure to Sb(III) at 1 ng/mL and 1 µg/mL compared to the control group (Figure 3D).

### 2.3. Mitochondrial Electron Leakage and ATP/ROS Ratio

Proton leakage was elevated in mitochondria treated with Sb(V) at 1 ng/mL and 1 µg/mL relative to the control group (Figure 4A). A similar increase in proton leakage was observed with Sb(III) at both doses (Figure 4B). The ATP/ROS ratio was reduced in mitochondria treated with Sb(V) at 1 ng/mL and 1 µg/mL compared to the control group (Figure 4C). The same trend was observed in Sb(III)-treated mitochondria, where the ATP/ROS ratio decreased at both dosages (Figure 4D).

### 2.4. Mitochondrial Transmembrane Potential (Δψm)

Treatment with Sb(V) resulted in a significant depolarization of mitochondrial transmembrane potential (Δψm) at both 1 ng/mL and 1 µg/mL doses compared to the control group (Figure 5A). Similarly, exposure to Sb(III) at the same doses caused a depolarization of Δψm relative to the control group (Figure 5B).

## 3. Discussion

To the best of our knowledge, the findings of this study demonstrate significant impairments in mitochondrial function caused by antimonials. Notably, these effects include a reduction in oxygen consumption capacity, particularly in the phosphorylative state, with decreased oxygen consumption observed in states 1, 2, and 3 of complex I-mediated respiration. This study represents the first detailed report on the direct interaction of antimonials with mammalian mitochondria and their subsequent impact on mitochondrial functionality.

Our data suggests a potential direct action of antimonials on the functioning of complex I, including during mitochondrial phosphorylative respiration. Mitochondrial dysfunction can arise from both exogenous and endogenous toxic insults [32]. Considering that the precise structural interactions of antimonial compounds remain poorly understood [7,11,24] and their mechanistic pathways are not fully characterized [7,11,33], it is plausible that these compounds interact with components of complex I, impairing its functionality. Sb(III) is classified as a borderline metal ion with a high affinity for ligands containing nitrogen and sulfhydryl groups [16]. It is hypothesized that its leishmanicidal mechanisms are linked to interactions with biomolecules containing sulfhydryl radicals or thiol groups, including thiols, peptides, proteins, and enzymes [11], and this effect cannot be excluded in mammalian cells. Previous studies have proposed that mitochondrial complex I is susceptible to oxidative thiol modifications [34], suggesting that the interaction of antimonials with mitochondrial thiol groups may contribute to its inhibitory effects, though this mechanism remains to be fully elucidated.

In this study, we observed alterations in mitochondrial ROS production at both 1 ng and 1 μg/mL doses of Sb(V), suggesting that the drug induces or amplifies its toxic effect via modulation of reactive oxygen species. It has been suggested that ROS production and the resulting oxidative stress may constitute the mechanism of Sb-induced DNA damage [35,36,37]. ROS are continuously produced by cell growth and eliminated by cellular antioxidant defenses, maintaining a stable level of ROS in living cells. Under certain conditions, the balance between ROS production and elimination shifts towards increased ROS production, leading to elevated levels of intracellular ROS and oxidative stress [35,36,37]. ROS are considered toxic by-products of aerobic metabolism and are implicated as key contributors to macromolecular damage [37]. Moreover, there is a well-established relationship between proton leak and ROS generation in the mitochondrial respiratory chain, with mitochondrial superoxide production being highly dependent on Δp in isolated mitochondria [37].

Exposure to Sb(V) and Sb(III) led to a reduction in ATP production. This finding is particularly significant given that ATP production and mitochondrial integrity are critical for cellular survival [38,39]. The observed reduction in ATP production may be associated with decreased oxygen consumption by complex I. If complex I function is compromised, this could result in a diminished proton gradient across the mitochondrial inner membrane, thereby impairing the activity of ATP synthase (complex V) and oxidative phosphorylation [40]. Consequently, the decline in ATP production is likely a direct outcome of complex I dysfunction [37].

Interestingly, it was possible to observe that electron leakage was greater in mitochondria incubated with the drug Sb(V), and the same occurred when mitochondria were exposed to the drug Sb(III). The process of electron transfer along complexes I, II, III, and IV inevitably leads to some electron leakage, which is then donated directly to oxygen. This results in the formation of highly reactive oxygen species, such as superoxide, which can damage phospholipids, proteins, and DNA inside the cell [41]. Under normal physiological conditions, cells can neutralize the harmful effects of ROS by various antioxidant defense systems. However, when the overproduction of ROS exceeds the capacity of these defense mechanisms, oxidative stress occurs, especially during aging, inflammation, and exposure to certain drugs. Since the main sources of ROS is complex I, complex III, and monoamine oxidase, other mitochondrial proteins and mtDNA near these sources of ROS are at risk of being damaged [41].

As for mitochondrial transmembrane potential (Δψm), it was possible to observe a significant increase in this potential when the mitochondria were exposed to the drug Sb(V), and a comparable pattern was observed when they were exposed to the drug Sb(III). The depolarization of Δψm with Sb(V) and Sb(III) exposure suggests heightened vulnerability to subsequent stressors. Δψm is a crucial indicator of mitochondrial health and functionality, reflecting the balance between electron transport chain activity and proton gradient maintenance [42]. Pathological depolarization of Δψm is often a hallmark of mitochondrial dysfunction, initiating a cascade of detrimental cellular events [42,43,44]. Furthermore, the loss of Δψm impairs energy-dependent cellular processes and can lead to increased production of ROS [44]. Additionally, mitochondrial depolarization is a key step in the activation of the mitochondrial permeability transition pore (mPTP), which can result in the release of pro-apoptotic factors like cytochrome c, initiating cell death pathways [42,43,44].

### Study Limitations

The results obtained in this study, although robust in their experimental context, should be interpreted with due consideration of their methodological limitations. The main limitation lies in the use of a model of mitochondria isolated from mouse hearts. Although this approach is a well-established technique, it simplifies the complex physiological environment of a living organism. Isolated mitochondria, devoid of the cellular and tissue interactions that occur in vivo, may behave differently in terms of their sensitivity and response to external agents [29,30]. To complement the assessment of mitochondrial function, there are other experimental models that could be explored; however, other models also have their limitations [31]. The model of mitochondrial isolation by differential centrifugation is the most well-established in our group. Additionally, direct incubation of isolated mitochondria with concentrations of 1 ng/mL and 1 µg/mL of pentavalent (Sb(V)) and trivalent (Sb(III)) antimony may lead to an overestimation or underestimation of the actual mitochondrial exposure that would occur in vivo, given that the bioavailability and pharmacokinetics of the drug influence the concentration that actually reaches the mitochondria of cardiomyocytes [36]. The use of a murine model is also an important limitation, as metabolic, physiological, and genetic differences between mice and humans mean that the results may not be directly transposable [37]. Finally, it is crucial to highlight that, although this study demonstrated that Sb(V) and Sb(III) exert direct and detrimental effects on mitochondria, disrupting key aspects of their functionality, and that these antimonials impair mitochondrial performance through interactions with complex I in the respiratory chain, leading to phosphorylation and increased oxidative stress, the precise molecular mechanisms by which these interactions occur have not yet been fully elucidated [29,30,31,36,37]. We observed how the drug is affecting mitochondrial function. Therefore, further studies are needed to better understand the actual damage caused by antimonials to mitochondria. Despite these limitations, the data presented provide important evidence of antimonial-induced cardiac mitochondrial dysfunction, and they do not invalidate the relevance and originality of the findings.

## 4. Materials and Method

### 4.1. Materials

All chemicals (analytical grade) were obtained from Sigma-Aldrich (St. Louis, MO, USA) if not otherwise specified. All solutions were freshly prepared and filtered (1.2 μm, Millipore, Burlington, MA, USA).

### 4.2. Animals and Ethical Approval

This study conformed to the Guide for the Care and Use of Laboratory Animals published by the US National Institute of Health (8th edition, 2011), and the experimental protocols were approved by the local Institutional Animal Care and Use Committee (protocol 119/21) approved in 16/02/2022. The animals were maintained in an animal room with controlled light (12:12 h light–dark cycle) and temperature (23–24 °C). The animals had ad libitum access to food and water. In this study, 10 male animals per group were used. The animals are C57BL/6J mice at 16 weeks.

In the acute analysis of the effects of Sb(V) and Sb(III), 10 animals were used. The same mitochondria isolated for the control group analysis were simultaneously used for evaluating exposure to Sb(V) and Sb(III) doses, as represented in Figure 6:

Control Group: Composed of fresh mitochondria from the hearts of healthy mice (CTRL); n = 10.

Sb(V) Group: Composed of fresh mitochondria from the hearts of healthy mice exposed to 1 µg/mL and 1 ng/mL doses of Pentavalent Sodium Antimoniate (SbV) at the time of the experiment; n = 10.

Sb(III) Group: Composed of fresh mitochondria from the hearts of healthy mice exposed to 1 µg/mL and 1 ng/mL doses of Trivalent Sodium Antimoniate (SbIII) at the time of the experiment; n = 10.

### 4.3. Mitochondria Isolation

The isolation of mitochondria followed previously established protocols [45,46,47]. Mice were euthanized, and a thoracotomy was performed to carefully extract the heart, which was immediately placed in a tube containing ice-cold isolation buffer (250 mmol/L sucrose, 10 mmol/L HEPES, and 1 mmol/L ethylene glycol tetra-acetic acid (EGTA), pH adjusted to 7.4 using TRIS solution) maintained at 4 °C. This step helped remove excess blood. To optimize the procedure, 0.5% (*w*/*v*) bovine serum albumin (BSA) was added to the buffer. Adipose tissue was meticulously removed using scissors. The heart tissue was then minced into small pieces, approximately 1–2 mm in size, ensuring that all visible fat was eliminated. The minced tissue was homogenized using an Ultra-Turrax tissue homogenizer (KA Labortechnik, Staufen, Germany). Homogenization was performed on ice for two 10 s cycles at 6500 rpm. The homogenate was further processed with a Potter–Elvehjem tissue homogenizer equipped with a Teflon pestle, aided by proteinase type XXIV (8 IU/mg tissue weight). The resulting homogenate was centrifuged at 700× *g* for 10 min at 4 °C. The supernatant was carefully collected and subjected to further centrifugation at 12,300× *g* for 10 min. The pellet obtained from this step was resuspended in ice-cold isolation buffer without BSA and centrifuged again at 10,300× *g* for 10 min at 4 °C. This process was repeated to ensure purity, and the final pellet was resuspended in isolation buffer. The protein concentration in the final mitochondrial pellet was quantified using the Lowry protein assay (Biorad, Hercules, CA, USA) with bovine serum albumin (BSA; Thermo Scientific, Waltham, MA, USA) as the standard.

### 4.4. Mitochondrial Oxygen Consumption Protocol

Mitochondrial respiration for complexes I (states 1, 2, and 3), IV and maximal uncoupling oxygen uptake were evaluated using a two-chamber respirometer. Measurements were performed with a Clark-type oxygen electrode (Strathkelvin, Glasgow, UK) at 37 °C under magnetic stirring in an incubation buffer. The buffer composition included 125 mmol/L KCl, 10 mmol/L MOPS, 2 mmol/L MgCl_2_, 5 mmol/L KH_2_PO_4_, and 0.2 mmol/L EGTA, with substrates such as glutamate (5 mmol/L) and malate (5 mmol/L) for complex I or succinate (5 mmol/L) for complex II. The oxygen electrode was calibrated using a solubility coefficient of 217 nmol O_2_/mL at 37 °C. To measure complex I respiration, mitochondria (containing 100 µg of mitochondrial protein) were added to 1 mL of the incubation buffer. After 2 min of equilibration, ADP (1 mmol/L) was introduced, and ADP-stimulated respiration was recorded over a 2 min interval. For complex II respiration, the complex I inhibitor rotenone (1 µmol/L) was added. Subsequent measurements of complex IV respiration and maximal uncoupled oxygen uptake were performed directly in the respiration chamber, or aliquots of the mitochondrial incubation buffer were taken to assess ATP production and mitochondrial ROS levels. Complex IV respiration was stimulated by adding N,N,N′,N′-tetramethyl-p-phenylenediamine (TMPD, 300 µmol/L) and ascorbate (3 µmol/L), which serve as electron donors to cytochrome oxidase by reducing cytochrome c. Maximal uncoupled oxygen consumption was determined in the presence of 30 nmol/L carbonyl cyanide-p-trifluoromethoxyphenylhydrazone (FCCP) [45,46,47].

### 4.5. Mitochondrial ATP Production

Following the measurement of ADP-stimulated respiration for complexes I and II, the mitochondrial incubation buffer was collected from the respiration chamber and promptly supplemented with a 1:5 dilution of the ATP assay mix (ATP Bioluminescence Assay Kit, Sigma-Aldrich, St. Louis, MO, USA). Mitochondrial ATP production was measured immediately after each respiration assessment. ATP levels were quantified by comparing the bioluminescence signal to a series of ATP standards using a 96-well white plate in a spectrofluorometer (SpectraMax^®^ M3, Molecular Devices, San Jose, CA, USA) with an emission wavelength of 560 nm [45,46,47].

### 4.6. Mitochondrial ROS Concentration

Mitochondrial reactive oxygen species (ROS) levels were measured using the Amplex Red Hydrogen Peroxide Assay Kit (Life Technologies, Carlsbad, CA, USA). Amplex Red reacts with hydrogen peroxide in a 1:1 stoichiometry in the presence of horseradish peroxidase (HRP), producing highly fluorescent resorufin with 95% efficiency. The mitochondrial incubation buffer was collected from the respiration chamber and immediately supplemented with 50 µmol/L Amplex UltraRed reagent and 2 U/mL Pierce™ Horseradish Peroxidase (HRP, Thermo Fisher, Catalog No. 31491, Waltham, MA, USA) [45,46,47].

### 4.7. Mitochondrial Transmembrane Potential

A mitochondrial suspension (100 μg/mL) was added to the incubation buffer in the absence of respiratory substrates at 37 °C and under constant stirring. The mitochondrial transmembrane potential was determined using the probe TMRM (tetramethylrhodamine methyl ester, 400 nmol·L^−1^) in the incubation solution containing 100 μg/mL of mitochondria for 1 h prior to the experiment. The transmembrane potential was estimated by fluorescence emitted by TMRM under 580 nm excitation. The transmembrane potential was expressed as the percentage of fluorescence emitted by TMRM-labeled mitochondria in the presence of cyclosporin A (0% of mitochondrial depolarization) relative to that emitted after the addition of FCCP to fully depolarize the mitochondria (100% of mitochondrial depolarization) [45,46,47].

### 4.8. Electron Leakage and ATP/ROS Production Ratio

Electron leakage is the loss of the electron from the electron transport chain to form superoxide (O_2_^−^). However, other reactive oxygen species, such as hydroperoxyl radical (HO_2_) and hydrogen peroxide (H_2_O_2_), might occur spontaneously (e.g., pH-dependent) or under the action of antioxidant enzymes (e.g., superoxide dismutase). The site of initial leakage is often considered to be a semiquinone radical (QH) or reduced flavin (FMN and FAD). To calculate the fraction of electrons leaked out of the respiratory chain, the H_2_O_2_ formation is divided by the rate of mitochondrial O_2_ consumption. Thus, we were able to determine the electron leakage inherent to ROS production. Here, electron leakage was calculated using the mitochondrial O_2_ consumption from complex I under state 3 stimulation and ROS production from the same complex and state. The ATP/ROS ratio should be measured to determine the formation of ROS linked to O2 consumption with ATP as a product [45,46,47].

### 4.9. Statistical Analysis

The sample size for each experiment was carefully determined through statistical power analysis using G*Power version 3.1.9.7 (Heinrich Heine University Düsseldorf, Düsseldorf, Germany). For every parameter measured, we estimated the effect size (Cohen’s f) based on the coefficient of determination (R^2^) from analysis of variance (ANOVA) performed on data. These values were then used to calculate the required sample size for a fixed-effects one-way ANOVA F test, with a significance level of α = 0.05 and a desired power of 80% (1 − β = 0.8). The minimum number of samples per group varied depending on the parameter and effect size observed, helping to ensure reliable results while avoiding unnecessary use of animals. For graphic and statistical analysis, the software GraphPad Prism 8.4.3 (San Diego, CA, USA) was used. The data distribution was considered normal by the Shapiro–Wilk test. The significant differences in mitochondrial functions were evaluated by the parametric one-way ANOVA test followed by the Bonferroni post hoc test. Data are presented as the mean ± standard deviation (SD). *p* < 0.05 was considered statistically significant.

## 5. Conclusions

The findings of this study reveal that Sb(V) and Sb(III) exert direct and detrimental effects on mitochondria, disrupting key aspects of their functionality. The data strongly suggest that these antimonials impair mitochondrial performance through interactions with complex I in the respiratory chain, leading to compromised oxidative phosphorylation and heightened oxidative stress, as demonstrated in Figure 7. This underscores the potential mechanistic role of complex I disruption in the mitochondrial toxicity of antimonials (Figure 7).

## Figures and Tables

**Figure 1 ijms-26-09073-f001:**
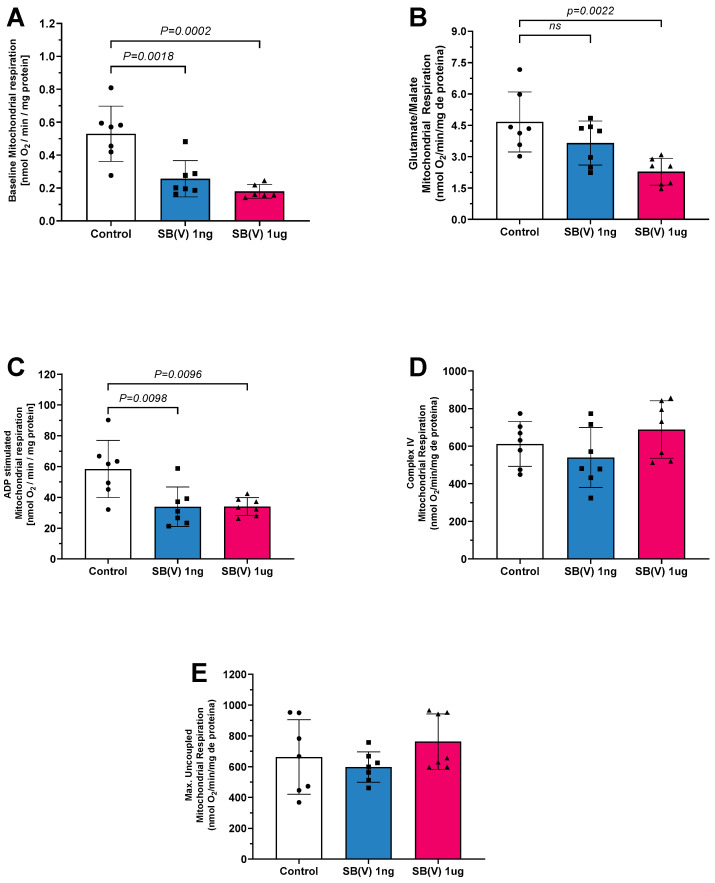
Mitochondrial respiration in the presence of Sb(V) at concentrations of 1 ng/mL and 1 µg/mL. (**A**) Baseline respiration (state 1 complex I), (**B**) Glutamate/malate stimulation (state 2 complex I) respiration, (**C**) ADP stimulation (state 3 complex I—phosphorylative state) respiration. (**D**) Complex IV respiration stimulated by TMPD and ascorbate, and (**E**) maximal uncoupled oxygen uptake induced by FCCP of isolated mitochondria from mice hearts. Each symbol represents one animal. Data are presented as the mean ± standard deviation (SD). The statistics were calculated by parametric one-way ANOVA. Horizontal square brackets indicate significant differences and the corresponding *p*-value.

**Figure 2 ijms-26-09073-f002:**
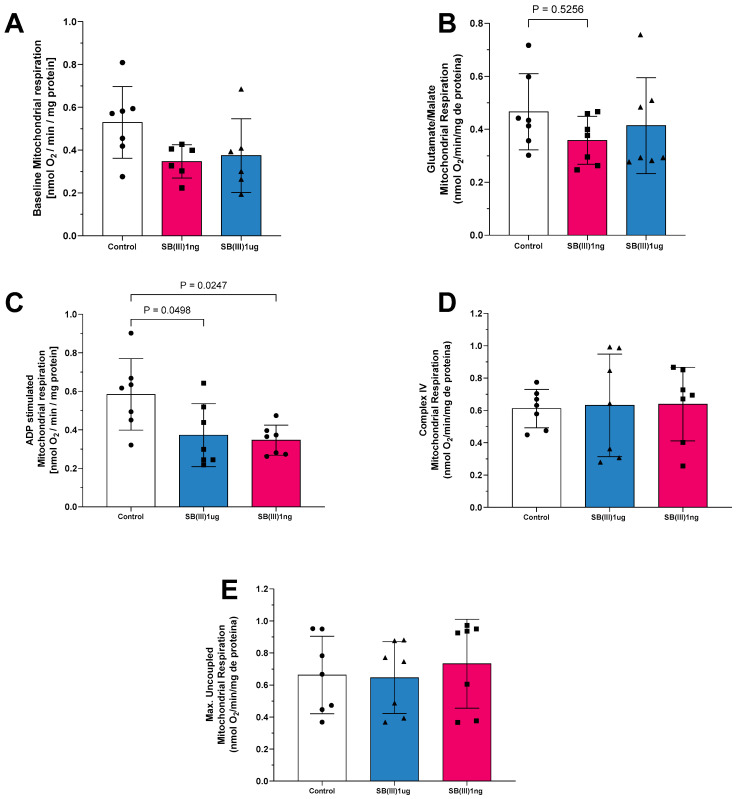
Mitochondrial respiration in the presence of Sb(III) at concentrations of 1 ng/mL and 1 µg/mL. (**A**) Baseline respiration (state 1 complex I). (**B**) Glutamate/malate stimulation (state 2 complex I) respiration. (**C**) Adenosine diphosphate (ADP) stimulation (state 3 complex I—phosphorylative state) respiration. (**D**) Complex IV respiration stimulated with N,N,N,N-tetramethyl-p-phenylenediamine dihydrochloride (TMPD) and ascorbate. (**E**) Maximal uncoupled oxygen uptake induced by carbonyl cyanide 4-(trifluoromethoxy)phenylhydrazone (FCCP) of isolated mitochondria from mice hearts. Each symbol represents one animal. Data are presented as the mean ± standard deviation (SD). The statistics were calculated by parametric one-way ANOVA. Horizontal square brackets indicate significant differences and the corresponding *p*-value.

**Figure 3 ijms-26-09073-f003:**
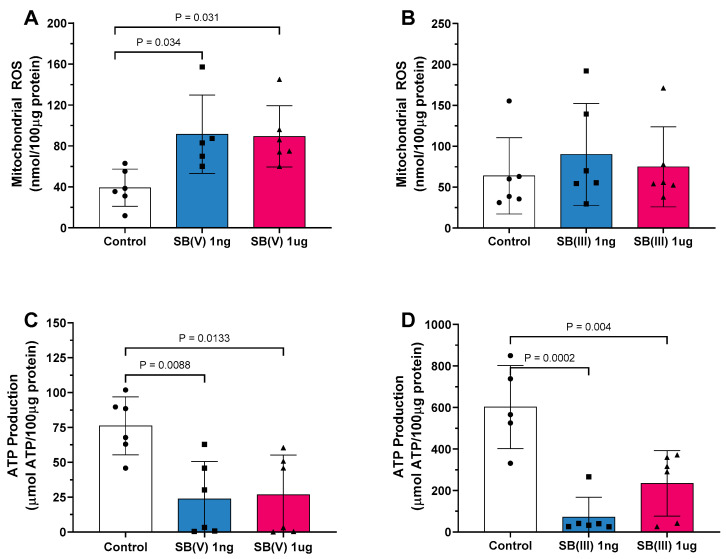
Mitochondrial products and characteristics. (**A**) Reactive oxygen species (ROS) production in the presence of Sb(V) at concentrations of 1 ng/mL and 1 µg/mL. (**B**) Reactive oxygen species (ROS) production in the presence of Sb(III) at concentrations of 1 ng/mL and 1 µg/mL. (**C**) Adenosine triphosphate (ATP) production in the presence of Sb(V) at concentrations of 1 ng/mL and 1 µg/mL. (**D**) Adenosine triphosphate (ATP) production in the presence of Sb(III) at concentrations of 1 ng/mL and 1 µg/mL Each symbol represents one animal. Data are presented as the mean ± standard deviation (SD). The statistics were calculated by parametric one-way ANOVA. Horizontal square brackets indicate significantly different and corresponding *p*-value.

**Figure 4 ijms-26-09073-f004:**
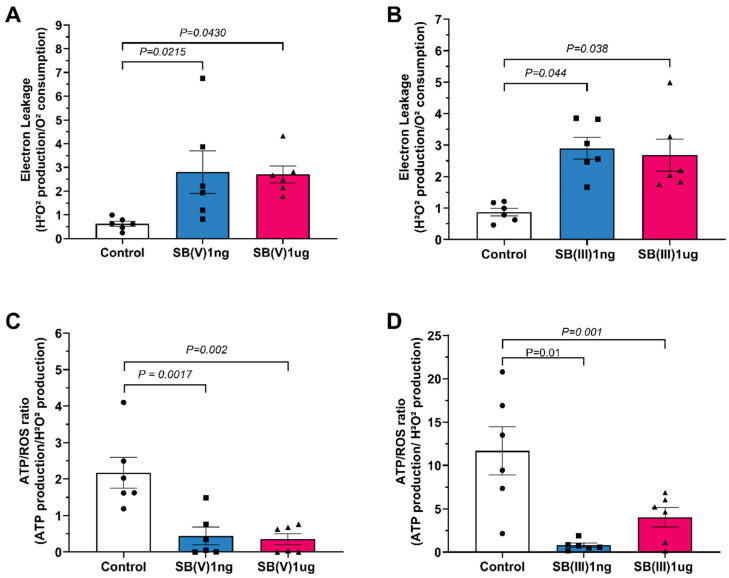
Mitochondrial products and characteristics. (**A**) Electron leakage production in the presence of Sb(V) at concentrations of 1 ng/mL and 1 µg/mL. (**B**) Electron leakage production in the presence of Sb(III) at concentrations of 1 ng/mL and 1 µg/mL. (**C**) ATP/ROS ratio production in the presence of Sb(V) at concentrations of 1 ng/mL and 1 µg/mL. (**D**) ATP/ROS ratio of Sb(III) at concentrations of 1 ng/mL and 1 µg/mL Each symbol represents one animal. Data are presented as the mean ± standard deviation (SD). The statistics were calculated by parametric one-way ANOVA. Horizontal square brackets indicate significantly different and corresponding *p*-value.

**Figure 5 ijms-26-09073-f005:**
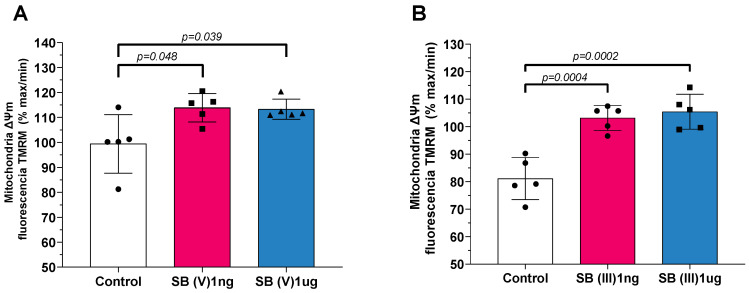
Mitochondrial products and characteristics. (**A**) Mitochondrial transmembrane potential (mΔψ) of isolated mitochondria from mice hearts in the presence of Sb(V) at concentrations of 1 ng/mL and 1 µg/mL. (**B**) Mitochondrial transmembrane potential (mΔψ) of isolated mitochondria from mice hearts in the presence of Sb(III) at concentrations of 1 ng/mL and 1 µg/mL. Each symbol represents one animal. Data are presented as the mean ± standard deviation (SD). The statistics were calculated by parametric one-way ANOVA. Horizontal square brackets indicate significantly different and corresponding *p*-value.

**Figure 6 ijms-26-09073-f006:**
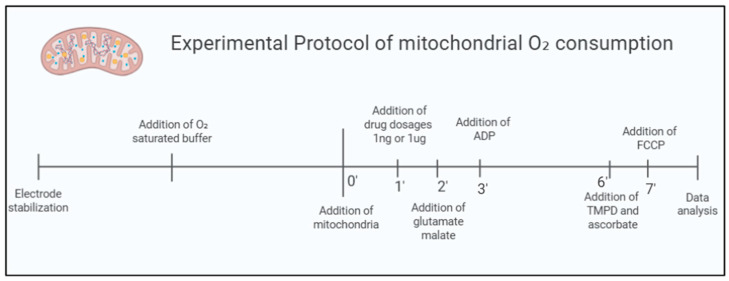
Experimental protocol for oxygen consumption applied to mitochondrial respiration was measured using a Clark-type electrode. The experimental protocol includes the following steps: electrode stabilization, addition of isolated mitochondria, addition of the tested drug (duration: 1 min), addition of glutamate and malate (to initiate substrate-driven respiration, duration: 1 min), addition of ADP (to activate phosphorylative respiration, duration: 3 min), addition of TMPD and ascorbate (as electron donors to cytochrome c, duration: 2 min), and addition of FCCP (to assess maximal uncoupled oxygen consumption, duration: 1 min).

**Figure 7 ijms-26-09073-f007:**
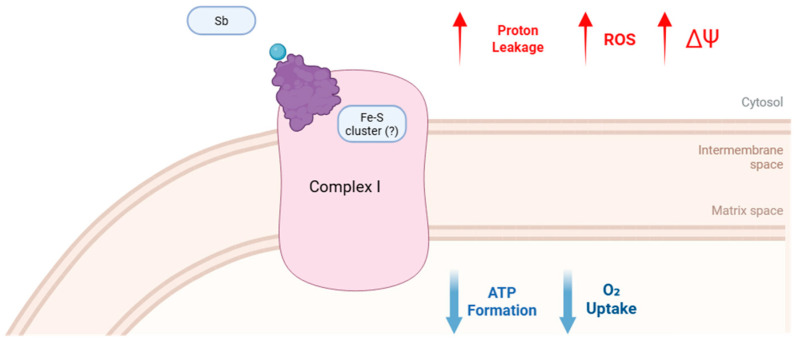
Sb interaction triggers mitochondrial impairment by disrupting electron transport. Schematic representation of the effects of Sb(V) and Sb(III) on mitochondrial complex I (?). Putatively Antimonials interact with Fe-S clusters, possibly by binding to the thiol (-SH) groups of cysteine residues, leading to complex I dysfunction. This interaction could result in decreased oxygen consumption and ATP synthesis, accompanied by increased proton leakage, enhanced production of reactive oxygen species (ROS), and alterations in mitochondrial membrane potential (ΔΨ). These changes culminate in impaired oxidative phosphorylation and exacerbated oxidative stress, highlighting complex I as a critical target of antimonial-induced mitochondrial toxicity. It should be noted, however, that the direct involvement of thiol groups in this process still requires further clarification.

## Data Availability

The data presented in this study are available on request from the corresponding author.

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
