# Peer review of "Trivalent and Pentavalent Antimonials Impair Cardiac Mitochondrial Function in Mice"

_ijms, 2025, doi:10.3390/ijms26189073_

Round 1

Reviewer 1 Report

Comments and Suggestions for Authors

The manuscript aim to elaborate the cardiotoxic effect of anti leishmanial treatment, Sb(V), through investigation of mitochondrial functions in mice cadiomyocytes  after treatment. This study provides the first report of impaired mammilian mitochondria after direct interaction with antimonial, however, all assays have been done in vitro. 

The manuscript is well-structured, clear, with various references ranging between old and new. the experimental design is well- defined with using of appropriate statistical methods. the results are reproducible based on the details given the the method section. The figures are clear and understandable with detailed legends. The conclusion is consistent with evidence and arguments presented. The study limitations were discussed and suggestions for future work were provided. 

 Minor comments:

1- I think there is no need to mention the statistical details in the abstract.

2-page 2 line 52, remove 7 after leishmaniasis. 

Author Response

Dear Reviewer
We sincerely thank the Reviewer for the careful evaluation of our manuscript and the constructive comments provided. All requested modifications have been made, including the removal of the statistical details from the abstract and the correction on page 2, line 52. In addition, the manuscript has been revised by a native English speaker to improve language clarity and fluency.

We greatly appreciate the Reviewer’s time and effort in assessing our work and for the valuable suggestions that have helped to enhance the scientific quality of the manuscript.

Reviewer 2 Report

Comments and Suggestions for Authors

The authors describe the toxicity mechanisms of Sb(V) and Sb(III) on the mitochondria from cardiac tissue in a murine model. The study is well-described with appropriate statistical analyses to support authors claims. I recommend publication with minor revisions.

  1. The manuscript describes, in words, how antimony may interact with enzymes. A graphic depicting their interaction would be helpful here to understand the thiophilic nature of antimonials. I believe this is an especially useful point to make because the focus on Complex I, which has more than 7 Fe-S clusters that may be affected by the introduction of antimony.
  2. Also, a graphic depicting the mechanism of toxicity would also be a helpful addition to the manuscript.

Author Response

Dear Reviewer,

We sincerely thank the Reviewer for the constructive comments and positive evaluation of our work. Following the suggestions, we have created a new figure, placed in the Conclusion section, that addresses both points raised: (i) the potential interaction of antimonials with thiol groups in Fe-S clusters, highlighting their thiophilic nature, and (ii) the proposed mechanism of antimony-induced mitochondrial toxicity, particularly at Complex I. In addition, a graphical abstract has also been included to further illustrate these processes in a clear and accessible way.

We believe these additions significantly enhance the clarity and impact of the manuscript.